# Evaluation of Congenital Cytomegalovirus Infection in Pregnant Women Admitted to a University Hospital in Istanbul

**DOI:** 10.3390/v16030414

**Published:** 2024-03-08

**Authors:** Evrim Ozdemir, Tugba Sarac Sivrikoz, Kutay Sarsar, Dilruba Tureli, Mustafa Onel, Mehmet Demirci, Gizem Yapar, Eray Yurtseven, Recep Has, Ali Agacfidan, Hayriye Kirkoyun Uysal

**Affiliations:** 1Department of Medical Microbiology, Istanbul Faculty of Medicine, Istanbul University, Istanbul 34093, Turkey; evrimozdemir2020@ogr.iu.edu.tr (E.O.); kutay.sarsar@istanbul.edu.tr (K.S.); onelm@istanbul.edu.tr (M.O.); gizem.yapar@ogr.iu.edu.tr (G.Y.); ali.agacfidan@istanbul.edu.tr (A.A.); 2Institute of Health Sciences, Istanbul University, Istanbul 34126, Turkey; 3Department of Obstetrics and Gynecology, Istanbul Faculty of Medicine, Istanbul University, Istanbul 34093, Turkey; tugba.ikoz@istanbul.edu.tr (T.S.S.); dilruba.tureli@istanbul.edu.tr (D.T.); recephas@istanbul.edu.tr (R.H.); 4Department of Medical Microbiology, Faculty of Medicine, Kirklareli University, Kirklareli 39100, Turkey; mdemirci1979@gmail.com; 5Department of Biostatistics, Istanbul Faculty of Medicine, Istanbul University, Istanbul 34093, Turkey; eyurt@istanbul.edu.tr

**Keywords:** CMV, prenatal infection, avidity, amniotic fluid, congenital CMV infections

## Abstract

Cytomegalovirus (CMV) can cause serious complications in immunocompromised individuals and fetuses with congenital infections. These can include neurodevelopmental impairments and congenital abnormalities in newborns. This paper emphasizes the importance of concurrently evaluating ultrasonography findings and laboratory parameters in diagnosing congenital CMV infection. To examine the prenatal characteristics of CMV DNA-positive patients, we assessed serum and amniotic fluid from 141 pregnant women aged 19–45 years, each with fetal anomalies. ELISA and PCR tests, conducted in response to these amniocentesis findings, were performed at an average gestational age of 25 weeks. Serological tests revealed that all 141 women were CMV IgG-positive, and 2 (1.41%) had low-avidity CMV IgG, suggesting a recent infection. CMV DNA was detected in 17 (12.05%) amniotic fluid samples using quantitative PCR. Of these, 82% exhibited central nervous system abnormalities. Given that most infections in pregnant women are undetectable and indicators non-specific, diagnosing primary CMV in pregnant women using clinical findings alone is challenging. We contend that serological tests should not be the sole means of diagnosing congenital CMV infection during pregnancy.

## 1. Introduction

Cytomegalovirus (CMV), part of the Betaherpesvirinae subfamily, enlarges virus-infected cells, hence its name (cyto: cell, mega: large) [1]. CMV, which has linear double-stranded DNA, consists of a phospholipid-rich envelope surrounding its icosahedral symmetric capsid [2,3,4]. It is globally prevalent and associated with periods of active viral replication, often detectable in body fluids and sometimes inducing symptoms similar to mononucleosis [5]. Antibodies are present in roughly 50–75% of adults in developed nations and over 90% in poorer regions or developing countries. Turkey has reported a CMV seropositivity of 96–99.8% [6,7,8,9,10].

Congenital CMV infections, usually caused by an active viral condition in the mother during pregnancy, can cross the placenta to the fetus [11]. Most babies born with congenital infections are asymptomatic, while about 5–10% are symptomatic [6,12]. Complications sometimes lead to early fatality in symptomatic infants (4%) but typically result in neurological defects, visual disturbances, cognitive impairment, and hearing loss [5,13].

Pregnancy prognosis can be deduced according to virus detection in amniotic fluid samples or CMV DNA after the initial 20–21 weeks [14,15]. Besides intrauterine infection, CMV can also occur during the pre and postnatal phases [16,17]. Placental transmission is necessary for infection persistence, but the virus can also spread through the cervix at birth or via breast milk postnatally [18,19]. Infants with symptoms are at a higher risk of complications compared to those without symptoms [20].

In congenitally infected infants, the most frequent neurological signs include microcephaly, sensorineural hearing loss (35%), hypotonia, lethargy, feeding difficulties, and seizures. About 10–15% may experience visual disturbances. Additional potential adverse outcomes are developmental disorders (40–50%) and premature labor (10–35%) [21].

The transmission rate of primary CMV infection, a condition that may have severe effects on the fetus, is 30–40%, with a higher risk in the third trimester. Specifically, the risk stands at 30% in the first trimester, 34–48% in the second trimester, and 40–72% in the third trimester. Despite this increasing risk, the most serious repercussions to the fetus occur in the first trimester [22]. For recurrent CMV infection, the chance of transmission from the mother to the fetus is considerably lower, at just 1–2% [22,23].

Different opinions exist on routine serological screening in CMV infections, making prenatal ultrasonography a crucial component of diagnosis. In particular, cranial findings are essential [24,25]. Detectable abnormalities can include intracranial calcifications, periventricular hyperechogenicity, an enlarged cisterna magna, ventriculomegaly, intraventricular synechiae, malformations in cortical development, thalamic hyperechogenicity, agenesis of the cerebellar vermis, and cerebellar cysts. Synechiae in the occipital horns of the lateral ventricle also serve as an important indicator. Accompanying symptoms of the infection may involve hepatosplenomegaly, intrauterine growth restriction (IUGR), hydrops fetalis, cardiomegaly, pericardial effusion, ascites, and a hyperechoic bowel [26]. With numerous variables, diagnosing congenital CMV infection prenatally proves challenging. The common diagnostic methods include serological tests, such as the anti-CMV IgG avidity index and molecular tests that identify CMV DNA in amniotic fluid samples. This study aims to detail the serological and molecular test results, along with other prenatal clinical factors, in a cohort of pregnant women with congenital anomalies identified using prenatal ultrasound.

## 2. Materials and Methods

The study cohort comprised 141 pregnant women, ranging from 19 to 45 years, who visited the Istanbul Faculty of Medicine’s Prenatal Diagnosis and Treatment Center due to sonographic indications of potential congenital infection. These patients, who had undergone extensive fetal anatomical screenings in our clinic, exhibited signs such as ventriculomegaly, ventricular synechiae, mega cisterna magna, abnormalities in the corpus callosum, polyhydramnios, microcephaly, an echogenic bowel, intrauterine growth retardation, and non-immune hydrops fetalis. Only those presenting these anomalies and screened for cytomegalovirus, excluding other congenital infections, were considered for the study. The inclusion criteria encompassed pregnant women with preliminary sonographic diagnoses of fetal anomalies and devoid of substance abuse, smoking, and alcohol consumption since their last menstrual period. Subjects were excluded if they demonstrated normal ultrasound results, had active sexually transmitted diseases, underwent artificial fertilization, or were in a consanguineous marriage.

On average, the ELISA and PCR tests were carried out at 25 weeks of gestation (between 22 and 32 weeks) in response to the findings of the amniocentesis. At the same time, we recommended fetal karyotyping and array CGH. If necessary, whole-exome sequencing was also suggested. This approach was deemed appropriate as fetal findings are a part of the differential diagnosis, regardless of other signs pointing toward a genetic disease risk, such as cortical dysplasia, congenital heart disease, and fetal growth restriction. It is noteworthy that no additional prenatal infections or genetic disorders were diagnosed in our study group aside from CMV DNA positivity, which signifies isolated prenatal CMV positivity.

We collected maternal serum and amniotic fluid samples simultaneously from pregnant women to detect CMV. These samples were promptly transported to the Department of Medical Microbiology laboratories at Istanbul Medical Faculty. We used the real-time PCR method to identify CMV DNA in the amniotic fluid. Additionally, we employed the ELISA method to detect CMV IgM, IgG, and IgG avidity in the maternal serum.

We tested the maternal serum samples using an automated Triturus ELISA instrument (Grifols, Madrid, Spain), following the manufacturer’s guidance. Commercial CMV ELISA IgM, IgG, and IgG avidity kits (Vircell, Granada, Spain) were used in the process. After birth, urine samples were collected and promptly sent to the laboratories at Istanbul Medical Faculty’s Department of Medical Microbiology.

We used the QIAsymphony SP-AS device (QIAGEN, Hilden, Germany) to extract DNA from the amniotic fluid samples for the quantitative detection of CMV DNA. The process involved utilizing the Artus^®^ CMV QS-RGQ Kit (QIAGEN, Hilden, Germany) and implementing the real-time PCR method in the Rotor-Gene Q device (QIAGEN, Hilden, Germany) according to the manufacturer’s guidelines. Moreover, the automated extraction of CMV DNA from 1200 μL amniotic fluid samples was carried out using the QIAsymphony DSP Virus/Pathogen Midi Kit (QIAGEN, Hilden, Germany).

The Artus^®^ CMV QS-RGQ kit is a commercially available kit for in vitro diagnostic use. It boasts an analytical sensitivity of 79.4 copies/mL and includes an internal control to identify false negatives. Each sample undergoes extraction after the addition of this internal control.

The description of the prenatal characteristics for CMV-positive cases was presented as a median, and frequency distributions were used for the data analysis.

The study adhered to the Declaration of Helsinki and received approval from the Ethics Committee of Istanbul University’s Faculty of Medicine. The approval was granted on 30 November 2021, under the protocol code 613315.

## 3. Results

Upon evaluating the serological test results, it was found that 141 pregnant women (100%) tested positive for CMV IgG, 139 (98.58%) tested negative for CMV IgM, and the CMV IgM test was suspected to be positive in 2 pregnant women (1.41%). In the samples with suspicion of a positive result, the CMV IgG avidity test indicated low avidity in the maternal serum. The analysis of the CMV DNA quantitative viral load results revealed positivity in 17 (12.05%) amniotic fluid samples from the pregnant women (Table 1). 

The average age of the 124 patients testing negative for CMV DNA was 31.80 years, while that of the 17 patients testing positive was 32.05 years. No statistically significant difference was observed between the two groups (*p* > 0.05). 

The previous childbirth history of these 17 mothers was 1.53 ± 1.17. Two (12%) of the positive cases resulted in fetal loss, seven (41%) resulted in pregnancy termination, and a live birth occured in eight (47%) of all cases. A total of 5 out of 8 of these children were boys and 3 were girls. The gestational age at delivery was 32 ± 3 weeks, and the types of delivery were 5 live births by caesarean section and 3 normal spontaneous births. The birth weight was 1.450 ± 432 g. The head circumference was 30.7 ± 1.2 cm. The type of feeding for 6 of these babies was breastfeeding, and 2 were given supplementary food.

Table 2 displays the results on CMV DNA, CMV IgG, CMV IgM, and CMV avidity. Confirmatory diagnoses for congenital CMV DNA were found in 8 out of 17 newborns based on the urine samples. Regarding the remaining 17, 9 terminations occurred due to severe findings, with no confirmations conducted. Tissue samples from the terminated fetuses could not be analyzed because the families, for religious reasons, did not permit sampling.

Table 3 demonstrates the prenatal characteristics and results of 17 cases where CMV PCR tests were positive following an amniocentesis (AC). Prenatal sonography detected these findings on average around the 23rd week of pregnancy (ranging from 22 to 29 weeks). The average gestational week for conducting the CMV PCR test due to these AC findings was approximately the 25th week (ranging from 22 to 32 weeks), which came out as positive. Central nervous system abnormalities showed up in 82% of cases with positive CMV PCR results, making them the most common prenatal findings. Notable among these were ventriculomegaly (47%), an abnormal corpus callosum (23%), and a significantly enlarged cisterna magna (17%). Other noticeable prenatal findings were IUGR (29%), a hyperechoic bowel (23%), and polyhydramnios (17%). Furthermore, 12% of the positive cases led to fetal loss, 41% ended in pregnancy termination, and live births made up 47% of the cases. Of these live births, one infant (12%) died on the second day due to a CMV-related pulmonary hemorrhage. In another case (12%), the infant was found to have sensorineural deafness in the postnatal hearing screening, after which a cochlear implant surgery was performed.

For the pregnant women with inconclusive maternal CMV serology results (CMV IgM suspicious positive, CMV IgG-positive), we performed CMV IgG avidity testing in the serum and CMV PCR after amniocentesis as a confirmation measure after prenatal ultrasonography and fetal evaluation. Following prenatal pediatric evaluations of the live births from pregnancies that tested negative for CMV DNA during amniocentesis, no clinical CMV symptoms were detected. We also examined any abnormal fetal findings like a hyperechogenic bowel, ventriculomegaly, IUGR, and corpus callosum abnormality, among others, for potential alternative causes. The pregnancies resulting in live births returned negative results for CMV DNA according to amniocentesis, while conventional karyotyping came out normal. In the instance of advised pregnancy termination, the contributing factors included cerebellar hypoplasia, severe ventriculomegaly, corpus callosum abnormality, and non-immune hydrops fetalis. 

Out of 141 pregnant women, all tested positive for CMV IgG antibodies, while 1.41% tested positive for CMV IgM antibodies. Low avidity was found in the maternal serum of the CMV IgM-positive samples. From the amniotic fluid samples of these women, 17 (12.05%) tested positive for CMV DNA. In 82% of cases with CMV PCR positivity detected using amniocentesis, an abnormality in the central nervous system was observed, making it the most common prenatal finding. Confirmatory diagnosis results for congenital CMV DNA from the urine samples after birth were positive in 8 out of 17 infants. Of the 17, 9 pregnancies were terminated due to severe findings. No confirmation was performed for the terminated pregnancies in our study.

## 4. Discussion

It should be noted that almost all infants showing clinical signs of congenital CMV infection are born to mothers who contracted a primary CMV infection during their pregnancy, and congenital CMV infection occurs in 1% of infants globally [25,26,27,28]. After 20 weeks of pregnancy, an ultrasound can highlight potential indications of congenital CMV infection, but these must be confirmed using serological or PCR tests [29]. The diagnosis of congenital CMV is complex and should not rely on serological tests alone. Therefore, our focus was on assessing serum and amniotic fluid samples for CMV derived from pregnant women who visited the obstetrics and gynecology outpatient clinics and were identified as having congenital anomalies. An additional focus was on identifying the prenatal characteristics in the PCR-positive patients.

Upon reviewing recent CMV seroprevalence studies from various cities in Turkey, it was found that CMV IgG seropositivity varied from 94% to 100% [27,28,30,31,32,33,34,35,36,37] (Table 4).

The variability in seropositivity has been linked to factors such as living conditions, economic and demographic characteristics, hygiene, and cultural and behavioral factors [32]. Our study found 100% CMV IgG seropositivity and 1.41% CMV IgM seropositivity, aligning with the figures reported in the literature. This suggests that Turkey’s average CMV IgG immunity is relatively high.

In a cohort study by Zuhair et al. [39], the CMV IgG seropositivity was found to be 18% in Canada, while it was discovered to be as high as 100% in a collective study spanning countries such as Egypt, Thailand, Iran, Nepal, and Turkey. The highest rate of seropositivity was found in our country at 97%, while the lowest rate was found in Ireland at 44%. Choi et al. [40] conducted a study involving 6837 women of childbearing age (15–49 years old), which revealed a CMV IgG seropositivity of 95.8% and a CMV IgM seropositivity of 2.4% among 5186 women in the same age bracket. Notably, the seropositivity was appreciably lower in women aged 15–20 (77.5%) as compared to women in older age groups. In our study of 141 pregnant women, the CMV IgG seropositivity rate was uniformly consistent across all age groups.

Symptomatic infants have a 40–60% chance of experiencing permanent sequelae, with sensorineural hearing loss being the most common [41]. In our sample, despite the small number of cases, it is notable that sensorineural deafness (SND) was identified in 12% of live-birth cases. This stresses the need for prenatal counseling for patients carrying fetuses impacted by the infection. According to documented studies and our current series, the most common prenatal CMV markers are central nervous system anomalies (82% of cases), with ventriculomegaly being the most common at 47% of cases. This is followed by intrauterine growth restriction, present in 29% of cases [42].

Opinions vary on the necessity of cytomegalovirus (CMV) screening during pregnancy. However, it is crucial for women of childbearing age, particularly those working in pediatric clinics or caring for children, to be screened for CMV before pregnancy. If found non-immune, risk reduction strategies include frequent handwashing after contact with the baby and monthly testing during the first half of the pregnancy. They should also be educated on how to lessen the risk of CMV infection during pregnancy and associated congenital diseases [43].

Many countries do not endorse regular screening for CMV in pregnant women for several reasons. These reasons include the high rate of CMV IgG seropositivity, increased false positivity in commercial IgM tests, and the potential of reactivation in a seropositive pregnant woman [44]. 

In CMV-infected fetuses, treatment of fetal CMV infection is important, and researchers suggest performing CMV screening in pregnancy for all pregnant women, even seropositive women, at about 8 and 14 weeks of pregnancy to evidence a possible primary or recurrent reactivation or reinfection, have an amniocentesis at 20 weeks and a Fetal Magnetic Resonance (FMR) scan if positive, and offer the choice of hyperimmune globulin and/or valaciclovir therapy, the termination of pregnancy, or continuing pregnancy under special controls and measures, including cesarian delivery [45,46,47,48].

There is recent evidence from a prospective, randomized, double-blind, placebo-controlled trial showing that valaciclovir therapy can reduce intrauterine CMV transmission after primary maternal infection [49]. The renal tubular epithelium is a key site for CMV replication. In the case of an infected fetus, the urinary excretion of CMV manifests while CMV DNA builds up in the amniotic fluid. The time it takes for fetal urinary excretion of CMV after maternal seroconversion or reactivation spans approximately 6–8 weeks [50].

Research has shown that there is no direct correlation between the viral load in the amniotic fluid and symptoms of prenatal CMV infection [29]. Using amniocentesis to test the viral load yields a sensitivity and specificity of 66.7% and 84.3%, respectively. This is particularly true for fetuses with no symptoms or without serious prenatal ultrasound deviations at the initial assessment. Thus, the ability of this testing method to predict symptomatic infection is somewhat limited. Consequently, in prenatal CMV infections, the severity of the accompanying prenatal symptoms holds more weight than the CMV DNA viral load detected using amniocentesis [29].

Pregnant women commonly undergo routine ultrasound scans around the 18–20-week mark to track fetal development. If the ultrasound detects any irregularities, it may suggest congenital cytomegalovirus (CMV) infection. In such scenarios, it is crucial to test the mother for primary CMV infection using serological evaluation and perform amniocentesis to check for fetal infection. Ultrasound abnormalities do not solely point to congenital CMV infection. However, confirming this viral infection plays a key role in evaluating potential risks and prognosis for the fetus [26]. For this study, prenatal sonography was conducted at 23 weeks gestation.

Fetal magnetic resonance (FMR) imaging acts as a supplementary tool to ultrasound for prenatal brain and body imaging. Observations such as ventriculomegaly, cortical malformations, calcifications, hepatosplenomegaly, liver signal alterations, and abnormal effusions are not exclusive to congenital CMV infection. FMR is also important for revealing cerebral abnormalities like polymicrogyria, lissencephaly, etc., which cannot be seen using ultrasound examination [51,52].

Our study’s limitations include its single-center design, CMV’s slow proliferation and latent features, and the sensitivity of the window period in patients with congenital anomalies. Other difficulties involve situations where serological tests are not conducted simultaneously and problems obtaining demographic data. Despite this, our data provide valuable insight for the combined evaluation of serological and viral load results from both amniocentesis and serum samples. We examined certain confounders that could cause anomalies, such as a latent virus being frequently reactivated or causing a reinfection in pregnancy, but we could not examine all factors, marking another limitation.

## 5. Conclusions

In conclusion, to anticipate the potential risk of a fetus’s CMV infection, especially in the first trimester, we recommend conducting maternal serological tests when the ultrasonographic findings indicate potential prenatal infections in the fetus or point to maternal contact/infection. However, serological tests being administered in the second trimester may not indicate maternal infections occurring early in the pregnancy. Furthermore, we stress the importance of the CMV IgG avidity test and CMV DNA PCR tests to differentiate acute from chronic disease and verify the infection when ultrasound detects anomalies in the fetus. Nevertheless, during periconceptional or early first-trimester maternal infection, high CMV IgG avidity may be observed at 20 weeks of pregnancy in CMV-seropositive mothers, making the identification of primary infection during pregnancy challenging. Infections occurring later in pregnancy are expected to have a substantially reduced impact on the fetus. We suggest performing CMV screening in pregnancy in all pregnant women, even seropositive women, at about 8 and 14 weeks of pregnancy.

## Figures and Tables

**Table 1 viruses-16-00414-t001:** Distribution of positivity of CMV IgM, CMV IgG, and the CMV IgG avidity tests in maternal blood samples and CMV DNA PCR tests in amniotic fluid samples by age group.

Age Groups	CMV IgM Positivity		CMV IgG Positivity		CMV IgG Avidity		CMV DNA			
							Negative		Positive	
	n	(%)	n	(%)	n	(%)	n	(%)	n	(%)
19–25 (n:26—18.44%)	-	0	26	100	-	0	25	96.1	1	3.9
26–30 (n:33—23.40%)	1 (suspected positive)	3	33	100	1 (low avidity)	3	26	78.7	7	21.3
31–36 (n:50—35.46%)	1 (suspected positive)	2	50	100	1 (low avidity)	2	44	88	6	12
37–40 (n:19—13.48%)	-	0	19	100	-	0	18	94.7	1	5.3
41–46 (n:13—9.22%)	-	0	13	100	-	0	11	84.6	2	15.4
Total (n:141)	2	1.41	141	100	2	1.41	124	87.9	17	12.05

**Table 2 viruses-16-00414-t002:** Distribution of CMV DNA, CMV IgG, CMV IgM, and CMV avidity.

No	Age	CMV DNA Conc—Amniotic Fluid Samples (Copies/mL)	CMV DNA Conc—Infant Urine Samples (Copies/mL)	CMV IgG (U./mL)	CMV IgM (U./mL)	CMV Avidity Index
1	29	94	191,486	Positive (200)	Negative	
2	27	54 *	285,587	Positive (200)	Negative	
3	34	43 *	342,674	Positive (200)	Negative	
4	41	21 *	ND	Positive (200)	Negative	
5	33	3,381,865	1,134,547	Positive (200)	Negative	
6	30	3,441,374	ND	Positive (200)	Negative	
7	36	5 *	ND	Positive (24.7)	Negative	
8	27	58,249,030	1,752,586	Positive (188)	Negative	
9	37	7 *	ND	Positive (200)	Negative	
10	36	9 *	ND	Positive (200)	Negative	
11	20	14 *	248,125	Positive (200)	Negative	
12	33	18 *	ND	Positive (200)	Negative	
13	29	4766	201,148	Positive (200)	Negative	
14	31	9 *	ND	Positive (127)	Negative	
15	45	82	ND	Positive (200)	Negative	
16	27	115	ND	Positive (200)	Suspicious (9.72)	Low avidity (0.15)
17	30	3,992,043	1,158,153	Positive (200)	Negative	
18	35	Negative	Negative	Positive (200)	Suspicious (9.97)	Low avidity (0.19)

* The value is under cut-off (79.4 copies/mL) value; ND: not done.

**Table 3 viruses-16-00414-t003:** Prenatal and perinatal characteristics of CMV PCR-positive cases (n = 17) in amniocentesis (AC).

Prenatal and Perinatal Characteristics	Median (Min–Max)	
Gestational week at prenatal diagnosis	23 (22–29)	
Gestational week at amniocentesis (AC)	25 (22–32)	
	n	%
Prenatal ultrasonographic features		
Central nervous system	14	82
Ventriculomegaly	8	47
Mega cisterna magna	3	17
Corpus callosum abnormality	4	23
Intraventricular synechia	2	12
Microcephalia	1	6
Periventricular echogenicity	2	12
Hyperechogenic intestine	4	23
Intrauterine growth restriction (IUGR)	5	29
Ascites	1	6
Polyhydramniosis	3	17
Perinatal outcome		
Intrauterine demise	2	12
Termination of pregnancy	7	41
Livebirth	8	47
Neonatal exitus *	1	12
Deafness ¶	1	12

AC: amniocentesis, CMV PCR: cytomegalovirus polymerase chain reaction, * postnatal second day due to pulmonary hemorrhage, ¶ operation for deafness, needed a cochlear implant.

**Table 4 viruses-16-00414-t004:** Distribution of CMV IgG, IgM seroprevalence, and the CMV IgG avidity data in some recent studies on pregnant women in different cities in Turkey.

Reference, City	CMV IgM Positivity (%)	CMV IgG Positivity (%)	CMV IgG Avidity Results
Bursal et al., 2021, Aydın [27]	2.6	98	N/A
Çubuk et al., 2020, Sivas [28]	0.7	99	N/A
Gülseren et al., 2019, Konya [30]	0.2	100	N/A
Obut et al., 2019, Diyarbakır [31]	0.7	99.2	N/A
Demir et al., 2019, İstanbul [32]	3.2	94	N/A
Altunal et al., 2018, İstanbul [33]	0.2 (Turkish citizens)	99.5 (Turkish citizens)	N/A
	0 (Syrian immigrants)	100 (Syrian immigrants)	N/A
Şirin et al., 2017, İzmir [34]	1.5	98.9	in IgM-positive 9 cases: Low avidity: 0 Intermediate: 1 High avidity: 8 (88.9%)
Şahiner et al., 2015, Ankara [35]	0.97	98.1	High avidity in all
Parlak et al., 2015, Van [36]	2.6	100	High avidity in all
Bakacak et al., 2014, Kahramanmaraş [37]	3.2	99.3	N/A
İnci et al., 2014, Artvin [38]	1.6	98.6	N/A
This study, 2022, Istanbul	1.41	100	Low avidity

N/A: not available.

## Data Availability

To protect patient confidentiality, the data used in the study cannot be shared with third parties.

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
