# Peer review of "Evaluation of Congenital Cytomegalovirus Infection in Pregnant Women Admitted to a University Hospital in Istanbul"

_viruses, 2024, doi:10.3390/v16030414_

Round 1

Reviewer 1 Report

Comments and Suggestions for Authors

In this study, a cohort of 141 pregnant women with fetuses affected by congenital anomalies was analyzed. Serological tests revealed that all were CMV IgG-positive, with 1.41% indicating recent infection. CMV DNA was detected in 12.05% of amniotic fluid samples, with 82% showing central nervous system abnormalities. The study concludes that maternal serological tests should be performed with ultrasonographic findings related to prenatal infections. It recommends the CMV IgG avidity test and CMV DNA PCR tests when ultrasound findings are present in the fetus, highlighting the importance of accurate diagnosis for assessing potential risks and prognosis.

This is a very descriptive study, but, given the lack of defined protocols in the management of congenital HCMV infections, new reports are always required.

To enhance the manuscript for publication, several improvements are suggested:

-          Provide more comprehensive data about the mothers, including their previous childbirth history, gestational age, and type of delivery. Include additional details about the newborns, such as sex, weight, head circumference, and type of feeding;

-          Statistical analysis is lacking throughout the manuscript;

-          Abstract, line 17 “141 pregnant women with congenital anomalies”: specify that fetuses have congenital anomalies. An overall editing of the abstract is encouraged, ensuring a clearer connection between the results presented and the study's conclusions.

-          Clarify the term “Salisarsara” in line 37, presumably correcting it to “saliva”.

-          Correct the capitalization of "High" to a lowercase letter in line 266.

-          Enhance the overall fluidity and readability of the manuscript by providing clearer explanations for interesting observations. Consider restructuring sentences and paragraphs for improved comprehension.

Comments on the Quality of English Language

The revision of the manuscript by a qualified English editing service is suggested.

Author Response

Dear Reviewer,

Thank you very much for your valuable comments and contribution. all comments have been carefully checked and edited on the manuscript.

You can also see our response below.

Thank you, Regards

Assoc. Prof. Hayriye Kirkoyun Uysal

Comments and Suggestions for Authors

In this study, a cohort of 141 pregnant women with fetuses affected by congenital anomalies was analyzed. Serological tests revealed that all were CMV IgG-positive, with 1.41% indicating recent infection. CMV DNA was detected in 12.05% of amniotic fluid samples, with 82% showing central nervous system abnormalities. The study concludes that maternal serological tests should be performed with ultrasonographic findings related to prenatal infections. It recommends the CMV IgG avidity test and CMV DNA PCR tests when ultrasound findings are present in the fetus, highlighting the importance of accurate diagnosis for assessing potential risks and prognosis.

This is a very descriptive study, but, given the lack of defined protocols in the management of congenital HCMV infections, new reports are always required.

To enhance the manuscript for publication, several improvements are suggested:

-          Provide more comprehensive data about the mothers, including their previous childbirth history, gestational age, and type of delivery. Include additional details about the newborns, such as sex, weight, head circumference, and type of feeding;

Response: Thank you very much for your valuable comments and contribution. this information has been added in the result section. “Previous childbirth history of these mothers was 1.53+1.17. Two (12%) of positive cases resulted in fetal loss, seven (41%) resulted in pregnancy termination, and a live birth occured in eight (47%) of all cases. These mothers age 5 out of 8 were boys and 3 were girls. Gestational age at delivery was 32+3 weeks and type of delivery was 5 of the live births were by caesarean section and 3 were normal spontaneous births. Birth weight was 1.450+432gr. Head circumference was 30.7+1.2 cm. Type of feeding of 6 of these babies were breastfeeding and 2 were supplementary food.”

-          Statistical analysis is lacking throughout the manuscript;

Response: Thank you very much for your valuable comments and contribution. Statistical comparison could only be made between age groups, but since this is not meaningful data, it is not included in the manuscript.

-          Abstract, line 17 “141 pregnant women with congenital anomalies”: specify that fetuses have congenital anomalies. An overall editing of the abstract is encouraged, ensuring a clearer connection between the results presented and the study's conclusions.

Response: Thank you very much for your valuable comments and contribution. Following sentence was changed to; “To determine the prenatal characteristics of CMV DNA positive patients, serum and amniotic fluid samples obtained from 141 pregnant women aged 19-45 years with congenital anomalies in the fetus were evaluated.” and after the American Manuscript editing service, these sentences changed to:

“To examine the prenatal characteristics of CMV DNA-positive patients, we assessed serum and amniotic fluid from 141 pregnant women aged 19–45 years, each with fetal anomalies.”

-          Clarify the term “Salisarsara” in line 37, presumably correcting it to “saliva”.

Response: Thank you very much for your valuable comments and contribution. The word saliva (as salisarsara) was misspelled. and after the American Manuscript editing service, these sentences changed to: “it is globally prevalent and associated with periods of active viral replication, often detectable in body fluids and sometimes inducing symptoms similar to mononucleosis.”

-          Correct the capitalization of "High" to a lowercase letter in line 266.

Response: Thank you very much for your valuable comments and contribution. Corrected.

-          Enhance the overall fluidity and readability of the manuscript by providing clearer explanations for interesting observations. Consider restructuring sentences and paragraphs for improved comprehension.

Response: Thank you very much for your valuable comments and contribution. The manuscript sent to American Manuscript editing service.you can see the editing certificate below.

Comments on the Quality of English Language

The revision of the manuscript by a qualified English editing service is suggested.

Reviewer 2 Report

Comments and Suggestions for Authors

Abstract line 15 spell out the abbreviation “USG”

Page 1 line 37 spelling error: “saliSarsara” = saliva

Page 2 line 47 suggestion: delete “multiplies and”

Page 2 line 61 suggestion: delete “Some infants with congenital CMV have symptoms at birth.” As it is repetition of line 47.

page 2 line 62 suggestion: change

“The effects of congenital CMV infection on the central nervous system are very important and the most common neurologic signs and symptoms in affected infants are…”

To

“The most common neurologic signs and symptoms in congenitally infected infants are…”

Page 2 line 86: suggestion: change “In our study, it was aimed” to “We aimed”

Page 2  line 87 and 88: the ages and location of the cohort may be described at the beginning of the methods section, not usually in the background or introduction section.

Page 2 lines 82 to 90: the reader does not get a sense of the gap in knowledge, and how the aims of the study fill that gap.

It does seem that the aim of the study should be to “describe” rather than to “evaluate” seeing that there is no control group (i.e. without congenital anomalies) against which the findings can be compared.

May I suggest rephrasing as follows:

“The prenatal clinical diagnosis of congenital CMV infection is difficult. Commonly available diagnostic tests include serological tests, including anti-CMV IgG avidity index and molecular tests detecting CMV DNA in amniotic fluid samples. We aimed to describe serological and molecular test results and other prenatal clinical characteristics of a cohort of pregnant women with congenital anomalies detected by prenatal ultrasound. “

Page 2 line 92: Suggestion: move the cohort description here:

“The cohort consisted of 141 pregnant women aged 19-45 years  who attended the Prenatal Diagnosis and Treatment Center at the Istanbul Faculty of Medicine, Department of Obstetrics and Gynecology and had fetal sonographic findings suggestive of congenital infection.”

Page 4 line 143 It is unclear to the reader what is meant by “suspicious positive samples”

Was CMV IgG avidity done on samples which were positive for IgM and IgG?

Page 4 line 150: it does seem that the CMV PCR positive rate of 21.3% in the 26-30yr age group is significantly above the average of 12.05% for all the age groups. Perhaps a statistician can assist here? It does seem biologically plausible that this age group would have a higher rate of CMV infections, given that they typically are on their second pregnancy and may be exposed to a CMV-shedding toddler in the household, resulting in reinfection during pregnancy. It is highly relevant to capture the parity of the cohort and how parity (and age of siblings) relates to CMV positivity in Table 1, for this reason.

Page 4  line 157: it is unclear to the reader what “questionable CMV IgM results” are. The word “suspicious was used in the table, and the word “questionable” is used in the text. Interpretation of test results should be defined clearly in the methods section and given consistent terminology e.g. “indeterminate” .

Page 4 line 160 “. During this period, the absence of viral infection in the amniotic fluid sample or a low viral load may explain the failure to detect CMV DNA.”

This is unlikely. If there was indeed fetal transmission ,the neonates urine sample at birth would not have been CMV PCR negative, as congenital transmissions typically shed in urine at high titres.

Page 5 Table 2 – suggest you specify in the heading of the column of urine results that it is infant Urine, as all other data in the table is maternal.

Page 5 table 2 –

It seems that there is a correlation between amniotic fluid titre and neonatal urine positivity.

Perhaps a statistician could assist here.

I wonder – in those patients in whom the amniotic fluid was positive but neonatal urine was negative – was it due to false positivity of the amniotic fluid CPR, or (more likely) due to an immunological difference in protection from vertical transmission – this may be explored further in the discussion if indeed a correlation is found.

Page 6  line 195 : it is unclear what “CMV IgM equivocal positive” means.

Page 6 line 196 “ we recommend…” this belongs in the discussion section not the results section. I don’t necessarily agree with this recommendation.

Looking at Table 2, it seems that patients 1, 2, 3, 5, 8, 11 and 13 were confirmed congenital infections (urine PCR positive at birth) which would have not received the benefit of avidity testing. I would suggest that perhaps avidity should be done on patients presenting with abnormal ultrasound findings and CMV IgG positive regardless of IgM result. IgM tests are notoriously insensitive.

Page 7 line 252 “SND was detected in 12% of live born cases” it would have been good to see these data in the results section rather than the discussion section.

Comments on the Quality of English Language

The English is understandable, besides a minor clarification as suggested in the comments above.

Author Response

Dear Reviewer,

Thank you very much for your valuable comments and contribution. all comments have been carefully checked and edited on the manuscript.

You can also see our response below.

Thank you, Regards

Assoc. Prof. Hayriye Kirkoyun Uysal

Comments and Suggestions for Authors

Abstract line 15 spell out the abbreviation “USG”

Response: Thank you very much for your valuable comments and contribution. Corrected.

Page 1 line 37 spelling error: “saliSarsara” = saliva

Response: Thank you very much for your valuable comments and contribution. The word saliva (as salisarsara) was misspelled. and after the American Manuscript editing service, these sentences changed to: “it is globally prevalent and associated with periods of active viral replication, often detectable in body fluids and sometimes inducing symptoms similar to mononucleosis.”

Page 2 line 47 suggestion: delete “multiplies and”

Response: Thank you very much for your valuable comments and contribution. Deleted.

Page 2 line 61 suggestion: delete “Some infants with congenital CMV have symptoms at birth.” As it is repetition of line 47.

Response: Thank you very much for your valuable comments and contribution. Deleted.

page 2 line 62 suggestion: change

“The effects of congenital CMV infection on the central nervous system are very important and the most common neurologic signs and symptoms in affected infants are…”

To

“The most common neurologic signs and symptoms in congenitally infected infants are…”

Response: Thank you very much for your valuable comments and contribution. Changed.

after the American Manuscript editing service, these sentences changed to: “In congenitally infected offspring, the most frequent neurological signs include microcephaly, sensorineural hearing loss (35%), hypotonia, lethargy, feeding difficulties, and seizures.”

Page 2 line 86: suggestion: change “In our study, it was aimed” to “We aimed”

Response: Thank you very much for your valuable comments and contribution. Changed.and after the American Manuscript editing service, these sentences changed to: “This study aims to…”

Page 2  line 87 and 88: the ages and location of the cohort may be described at the beginning of the methods section, not usually in the background or introduction section.

Response: Thank you very much for your valuable comments and contribution. This suggestion has been corrected with the following paragraph.

Page 2 lines 82 to 90: the reader does not get a sense of the gap in knowledge, and how the aims of the study fill that gap.

It does seem that the aim of the study should be to “describe” rather than to “evaluate” seeing that there is no control group (i.e. without congenital anomalies) against which the findings can be compared.

May I suggest rephrasing as follows:

“The prenatal clinical diagnosis of congenital CMV infection is difficult. Commonly available diagnostic tests include serological tests, including anti-CMV IgG avidity index and molecular tests detecting CMV DNA in amniotic fluid samples. We aimed to describe serological and molecular test results and other prenatal clinical characteristics of a cohort of pregnant women with congenital anomalies detected by prenatal ultrasound. “

Response: Thank you very much for your valuable comments and contribution. This suggestion has been corrected with the following paragraph.

Changed.and after the American Manuscript editing service, these sentences changed to: “With numerous variables, diagnosing congenital CMV infection prenatally proves challenging. Common diagnostic methods include serological tests, such as the anti-CMV IgG avidity index and molecular tests that identify CMV DNA in amniotic fluid samples. This study aims to detail the serological and molecular test results, along with other prenatal clinical factors, in a cohort of pregnant women with congenital anomalies identified by prenatal ultrasound.”

Page 2 line 92: Suggestion: move the cohort description here:

“The cohort consisted of 141 pregnant women aged 19-45 years  who attended the Prenatal Diagnosis and Treatment Center at the Istanbul Faculty of Medicine, Department of Obstetrics and Gynecology and had fetal sonographic findings suggestive of congenital infection.”

Response: Thank you very much for your valuable comments and contribution. This suggestion has been corrected with the following paragraph.

Changed.and after the American Manuscript editing service, these sentences changed to: “The study cohort comprised 141 pregnant women, ranging from 19 to 45 years, who visited the Istanbul Faculty of Medicine’s Prenatal Diagnosis and Treatment Center due to sonographic indications of potential congenital infection.”

Page 4 line 143 It is unclear to the reader what is meant by “suspicious positive samples”

Was CMV IgG avidity done on samples which were positive for IgM and IgG?

Response: Thank you very much for your valuable comments and contribution. An evaluation was made in accordance with the kit package insert (CYTOMEGALOVIRUS ELISA IgM Capture M1004), and since the IgM values ​​of two patients remained within the range of 9-11, they were considered suspiciously positive. Values ​​are in the CMV IgM column in table 2. Since the word "equvocal" is not commonly used, the word "suspicious" was chosen to make the meaning clear.

Page 4 line 150: it does seem that the CMV PCR positive rate of 21.3% in the 26-30yr age group is significantly above the average of 12.05% for all the age groups. Perhaps a statistician can assist here? It does seem biologically plausible that this age group would have a higher rate of CMV infections, given that they typically are on their second pregnancy and may be exposed to a CMV-shedding toddler in the household, resulting in reinfection during pregnancy. It is highly relevant to capture the parity of the cohort and how parity (and age of siblings) relates to CMV positivity in Table 1, for this reason.

Response: Thank you very much for your valuable comments and contribution. you have drawn attention to a very important epidemiological point. Unfortunately, we do not have these data you suggest. We will take this suggestion into consideration in our future studies.

Page 4  line 157: it is unclear to the reader what “questionable CMV IgM results” are. The word “suspicious was used in the table, and the word “questionable” is used in the text. Interpretation of test results should be defined clearly in the methods section and given consistent terminology e.g. “indeterminate” .

Response: Thank you very much for your valuable comments and contribution. The reason for being called suspicious is explained above. The word "questinable" has been replaced with "suspicious" to ensure text integrity.

Page 4 line 160 “. During this period, the absence of viral infection in the amniotic fluid sample or a low viral load may explain the failure to detect CMV DNA.”

 This is unlikely. If there was indeed fetal transmission ,the neonates urine sample at birth would not have been CMV PCR negative, as congenital transmissions typically shed in urine at high titres.

 Response: Thank you very much for your valuable comments and contribution. We have expressed this relationship incorrectly. The sentence has been removed in line with your suggestion.

Page 5 Table 2 – suggest you specify in the heading of the column of urine results that it is infant Urine, as all other data in the table is maternal.

Response: Thank you very much for your valuable comments and contribution. The word “infant” was added to table.

Page 5 table 2 –????

It seems that there is a correlation between amniotic fluid titre and neonatal urine positivity.

Perhaps a statistician could assist here.

I wonder – in those patients in whom the amniotic fluid was positive but neonatal urine was negative – was it due to false positivity of the amniotic fluid CPR, or (more likely) due to an immunological difference in protection from vertical transmission – this may be explored further in the discussion if indeed a correlation is found.

Page 6  line 195 : it is unclear what “CMV IgM equivocal positive” means.

Response: Thank you very much for your valuable comments and contribution. The word “equivocal” was changed to “suspicious”.

Page 6 line 196 “ we recommend…” this belongs in the discussion section not the results section. I don’t necessarily agree with this recommendation.

Looking at Table 2, it seems that patients 1, 2, 3, 5, 8, 11 and 13 were confirmed congenital infections (urine PCR positive at birth) which would have not received the benefit of avidity testing. I would suggest that perhaps avidity should be done on patients presenting with abnormal ultrasound findings and CMV IgG positive regardless of IgM result. IgM tests are notoriously insensitive.

Response: Thank you very much for your valuable comments and contribution. Following sentence was changed:

“For pregnant women with suspicious maternal CMV serology (CMV IgM suspicious positive, CMV IgG positive), were performed CMV IgG avidity testing in serum and CMV PCR at amniocentesis for confirmation after prenatal ultrasonography and fetal evaluation.”

Changed.and after the American Manuscript editing service, these sentences changed to: “..we performed…”

Page 7 line 252 “SND was detected in 12% of live born cases” it would have been good to see these data in the results section rather than the discussion section.

Response: Thank you very much for your valuable comments and contribution.  The sentence you want is between lines 188-190. Percentage data has been added.

Comments on the Quality of English Language

The English is understandable, besides a minor clarification as suggested in the comments above.

Response: Thank you very much for your valuable comments and contribution. The manuscript also sent to American Manuscript editing service.you can see the editing certificate below.

Round 2

Reviewer 1 Report

Comments and Suggestions for Authors

The authors answered properly to the criticisms. The paper can be accepted for publication.

Author Response

Thank you

Reviewer 2 Report

Comments and Suggestions for Authors

Suggestions have been adequately addressed

Author Response

Thank you
